# GUARD VECTOR: BEYOND ENGLISH LLM GUARDRAILS WITH TASK-VECTOR COMPOSITION AND STREAMING-AWARE PREFIX SFT

## ABSTRACT

We introduce *Guard Vector*, a safety task vector computed as the parameter difference between a guardrail model (Guard Model) and a same-architecture pretrained language model. Composing this vector with a target language model yields a *Target Guard Model (TGM)*. We then adapt TGM with a streaming-aware approach that combines *prefix-based training* and evaluation with a classifier that produces a *single-token output*. With this composition alone, TGM improves classification quality over established Guard Models across standard safety suites and enables language extensibility to Chinese, Japanese, and Korean, requiring neither additional training nor target language labels. It also demonstrates model portability across two widely used public guardrail backbones, Llama and Gemma. With prefix SFT (supervised fine-tuning), TGM preserves classification quality under streaming by aligning the behavior between prefix inputs and full-text inputs. The single-token output design increases throughput and reduces latency. Together, these components reduce data and compute requirements while promoting streaming-aware evaluation practices, thereby contributing to a more responsible AI ecosystem.

## 1 INTRODUCTION

Large language models (LLMs) are increasingly deployed across real-world applications, including online search engines, counseling, software development, finance, healthcare, and law (Mialon et al., 2023; Qu et al., 2025). This widespread application has increased demand for safety, reflecting concerns about the risks of incorporating LLMs in these sensitive domains. The predominant approach to ensuring LLM-safety is safety alignment, yet it suffers from inherent limitations such as vulnerability to novel jailbreaks, limited coverage of unseen risk categories, and inconsistent performance in cross-lingual settings (Bai et al., 2022). These shortcomings have motivated the use of guardrails as a complementary safeguard. Guardrails are specialized safety layers that classify or block model inputs and outputs based on predefined risk policies and have emerged as a practical mechanism for enforcing such safeguards (Markov et al., 2022; Ghosh et al., 2025).

However, implementing guardrails for non-English languages remains challenging due to the core reliance on English-centric models and policies (Llama Team, 2024; Team et al., 2024), the high cost of alignment pipelines that rely on supervised fine-tuning (SFT) and additional training (Zeng et al., 2024a; Han et al., 2024), and the limited availability of labeled datasets in target languages (Costa-Jussà et al., 2022). Another major challenge arises from streaming interactions, which are the default mode in many production environments. In streaming mode, chat LLM generates responses token by token, making it crucial for guardrails to provide immediate feedback and detect risk signals at early stages, especially for long outputs. Yet most guardrail research has focused on offline evaluation with access to full-text (Llama Team, 2024; Zeng et al., 2024a), with little systematic verification of standardized streaming metrics or parity with offline performance. As a result, ensuring accurate, high-throughput and low-latency guardrail decisions during streaming conditions remains an open challenge.

To address these limitations in non-English guardrail development, we are the first to propose **Guard Vector**, a **task-vector composition** method that transfers safety behaviors to target language models

Figure 1: Pipeline of task-vector composition and streaming adaptation. A *Guard Vector* is computed as the parameter difference between a Guard Model and a pretrained language model (PLM) of the same architecture. This vector is composed with a continual pretraining model (CP Model) in the target language to yield the Target Guard Model (TGM). Finally, streaming-aware prefix SFT with a single-token classifier aligns prefix and full-text behavior.

using publicly available weights and requiring neither additional training nor target language labels. Specifically, a *Guard Model* is a pretrained LLM that is fine-tuned for safety classification (i.e., guardrail) while a *pretrained language model (PLM)* is the same-architecture model without safety fine-tuning. The parameter difference between the Guard Model and PLM yields a *Guard Vector*. We then compose this resulting Guard Vector with a *continual pretraining model (CP Model)*, a same-architecture model further pretrained on large-scale non-English corpora, to obtain a *Target Guard Model (TGM)* with safety behaviors transferred to the target language. To address streaming evaluation and efficiency, we further introduce Target Guard Model with **streaming-aware prefix SFT (TGM (prefix SFT))**, a variant of TGM fine-tuned with cumulative prefixes and a single-token output classifier. This adaptation enables early detection from partial inputs while maintaining parity with offline evaluation and reducing latency through single-step classification. Figure 1 summarizes the complete pipeline from a given Guard Model, PLM, Guard Vector, CP Model, TGM, to TGM (prefix SFT).

We first compare the baseline Guard Model and TGM under the same architecture and observe that composition alone improves classification quality compared to the baseline. Next, we demonstrate portability of the method across both Llama and Gemma architectures using the same composition procedure. With the addition of prefix SFT, TGM further surpasses both the baseline's prefix SFT variant and the baseline Guard Model. Finally, we evaluate TGM under both offline (full-text) and streaming (prefix) regimes, reporting standardized streaming metrics and demonstrating parity with offline evaluation.

**The contributions of this paper are summarized as follows:**

- **First to address two practical gaps in guardrail deployment.** We demonstrate that (a) safety behaviors can be transferred to target language models *without additional training or target language labels* via composition of public weights, and (b) streaming guardrails can be trained and evaluated for *parity with offline* through a standardized prefix-based protocol with a single-token classifier.

- **Guard Vector: task-vector composition from public weights.** We define the Guard Vector as the parameter difference between a Guard Model and a same-architecture PLM, and compose it with a target language CP Model to obtain a TGM — requiring no additional training or labels. This enables portability across Llama and Gemma families and extensibility to Chinese, Japanese, and Korean.

- **Streaming-aware prefix SFT for low-latency guardrails.** We propose a prefix-based training and evaluation recipe where labels are inherited from full text, decisions are applied to cumulative prefixes, and inferences are used as a single-token classifier. TGM (prefix SFT) achieves offline–streaming parity while improving throughput and latency.

- **Responsible AI impact.** Our pipeline provides a training-light path to reliable non-English guardrails from public weights, reducing data and compute burden while promoting streaming-aware evaluation practices.

## 2 RELATED WORK

**Guardrail Models**  Open-weight guardrails generally follow a *backbone + supervised fine-tuning (SFT) or low-rank adaptation (LoRA)* recipe on English-centric data. Llama Guard 3 and 4 apply SFT on Llama backbones, with language coverage centered on English and only limited non-English reporting (Llama Team, 2024; 2025). NeMoGuard (AEGIS 2.0) releases LoRA adapters on Llama-3.1-Instruct; despite broad safety categories, its dataset primarily targets English, limiting applicability in non-English contexts (Ghosh et al., 2025; Llama Team, 2024). ShieldGemma is built on Gemma-2, defines the problem in English and aggregates per-category judgments at inference, which can add latency relative to single-pass binary classification (Zeng et al., 2024a; Team et al., 2024). A compact landscape is summarized in Table 1.

**Further related work.**  See Appendix F for task-vector arithmetic, streaming-aware guardrails, and evaluation-protocol details.

| Model | BB | NT | CJK |
|---|---|---|---|
| Llama Guard 3 | L | ✗ | ✗ |
| Llama Guard 4 | L | ✗ | ✗ |
| NeMoGuard | L | ✗ | ✗ |
| ShieldGemma | G | ✗ | ✗ |
| Target Guard Model | L/G | ✓ | ✓ |

Table 1: Comparison across backbone (BB; L=Llama, G=Gemma), need for additional training (NT; ✓ means none), and Chinese-/Japanese/Korean coverage (CJK). Target Guard Model spans both backbones and is the only entry that requires no additional training while demonstrating non-English coverage.

## 3 METHODOLOGY

We compute a safety task vector (Guard Vector) as the parameter difference between a guardrail model (Guard Model) and a pretrained language model (PLM) of the same architecture. Given a target language continual pretraining model (CP Model) with the same architecture, we compose the Guard Vector with the CP model to obtain a Target Guard Model (TGM). To support streaming regimes, we introduce a streaming-aware prefix SFT recipe that supervises decisions on cumulative prefixes and uses a single-token output classifier to align prefix and full-text behavior while improving efficiency.

### 3.1 GUARD VECTOR COMPOSITION

**Notation and assumptions.**  Assume all models share the same architecture. Let $\theta_{\mathrm{PLM}}, \theta_{\mathrm{GM}}, \theta_{\mathrm{CP}}$ denote the parameters of PLM, GM, and the target language CP Model, respectively. Each parameter set $\theta$ is a map from parameter names to tensors; write $\mathrm{keys}(\theta)$ for its key set and index tensors by $t$ (e.g., $\theta[t]$). We define the excluded parameter types and the composition domain as

$$\mathcal{E} = \{\text{embeddings, lm\_head, LayerNorm}\}, \ S = \big(\mathrm{keys}(\theta_{\mathrm{PLM}}) \cap \mathrm{keys}(\theta_{\mathrm{GM}}) \cap \mathrm{keys}(\theta_{\mathrm{CP}})\big) \setminus \mathcal{E}. \quad (1)$$

with an additional requirement of exact tensor-shape equality; if shapes mismatch for a key, that key is not included in $S$. We write $\theta_{\mathrm{TGM}}$ for the composed model and $\theta_{\mathrm{TGM\text{-}SFT}}$ for its prefix-SFT adaptation. Labels are binary, $y \in \{\text{SAFE, UNSAFE}\}$, mapped to $\{0, 1\}$ with UNSAFE$= 1$.

**Guard Vector and composition rules.**  For each $t \in S$, define the Guard Vector as the parameter difference

$$V_{\mathrm{GV}}[t] = \theta_{\mathrm{GM}}[t] - \theta_{\mathrm{PLM}}[t]. \quad (2)$$

Obtain the **Target Guard Model (TGM)** by adding the Guard Vector to the CP Model:

$$\theta_{\mathrm{TGM}}[t] = \theta_{\mathrm{CP}}[t] + V_{\mathrm{GV}}[t], \quad t \in S; \qquad \theta_{\mathrm{TGM}}[t] = \theta_{\mathrm{CP}}[t], \quad t \notin S. \quad (3)$$

---

**Algorithm 1** Compose Target Guard Model with Guard Vector

---

**Require:** $\theta_{\text{PLM}}$ (e.g., Llama 3.1), $\theta_{\text{GM}}$ (e.g., Llama Guard 3), $\theta_{\text{CP}}$ (target language CP Model); same architecture

**Ensure:** $\theta_{\text{TGM}}$
 1: $\mathcal{E} \leftarrow \{\text{embeddings, lm\_head, LayerNorm parameters}\}$
 2: $S \leftarrow \big(\text{keys}(\theta_{\text{PLM}}) \cap \text{keys}(\theta_{\text{GM}}) \cap \text{keys}(\theta_{\text{CP}})\big) \setminus \mathcal{E}$
 3: **for** each $t \in S$ **do**
 4: $\quad V_{\text{GV}}[t] \leftarrow \theta_{\text{GM}}[t] - \theta_{\text{PLM}}[t]$
 5: $\quad \theta_{\text{TGM}}[t] \leftarrow \theta_{\text{CP}}[t] + V_{\text{GV}}[t]$
 6: **end for**
 7: **for** each $t \notin S$ **do**
 8: $\quad \theta_{\text{TGM}}[t] \leftarrow \theta_{\text{CP}}[t]$
 9: **end for**
10: **return** $\theta_{\text{TGM}}$

---

No scaling factor is applied. A summary of the procedure appears in Algorithm 1.

### 3.2 TARGET GUARD MODEL (PREFIX SFT): STREAMING-AWARE PREFIX TRAINING AND LATENCY-AWARE DESIGN

**Streaming-aware prefix training.** We propose **prefix SFT** for $\theta_{\text{TGM}}$ to support streaming deployment. For a model-generated response $r$ of length $L$, we construct cumulative prefixes under a monotone prefix schedule $\mathcal{K}(r) \subseteq \{1, \dots, L\}$:

$$\mathcal{C}(r) \;=\; \{\, r_{1:K} \mid K \in \mathcal{K}(r) \,\}. \tag{4}$$

This schedule can be defined at different granularities (characters, tokens, or sentence boundaries). Unless otherwise noted, we instantiate a character-based schedule $\mathcal{K}(r) = \{100, 200, \dots, L\}$: it is tokenizer-agnostic (avoids mismatches between the generator and the guardrail), robust to adversarially long outputs, and reduces bookkeeping and context-tracking overhead.

Each prefix $r_{1:K} \in \mathcal{C}(r)$ inherits the label of the full response. If the original is SAFE, all prefixes are SAFE; if the original is UNSAFE, prefixes strictly before the first occurrence of harmful content are SAFE, and prefixes at or after that point are UNSAFE. We discard sequences that violate monotonicity (e.g., SAFE→UNSAFE→SAFE) and UNSAFE cases with no detected harmful prefix. This supervision targets early detection under streaming regimes.

**Prefix-level class balancing.** The $\mathcal{C}(\cdot)$ expansion alters class balance at the prefix level: when UNSAFE responses tend to be longer, more prefixes are labeled UNSAFE, while shorter SAFE responses contribute fewer prefixes. To prevent this drift from biasing training, we rebalance the prefix pool to a 1:1 SAFE/UNSAFE ratio by downsampling the majority class at the prefix level. We considered class-weighted losses as an alternative, but found simple resampling sufficient in our setting. Detailed training dataset counts and resampling settings are provided in Appendix B.

**Single-token classification.** We classify with a single-token output: concretely, we use the model's next-token distribution restricted to two reserved label tokens $\mathcal{V}_y = \{v_{\text{<SAFE>}}, v_{\text{<UNSAFE>}}\}$ added to the tokenizer. Given a prefix prompt $p(r_{1:K})$, let $z_{\text{SAFE}}$ and $z_{\text{UNSAFE}}$ denote the next-token logits for these two labels, and define

$$p_\theta(\text{UNSAFE} \mid p(r_{1:K})) \;=\; \frac{\exp(z_{\text{UNSAFE}})}{\exp(z_{\text{SAFE}}) + \exp(z_{\text{UNSAFE}})}. \tag{5}$$

We train with binary cross-entropy (UNSAFE= 1):

$$\mathcal{L}(\theta) = - \sum_{(r_{1:K}, \, y)} \Big[ y \log p_\theta(\text{UNSAFE} \mid p(r_{1:K})) + (1 - y) \log \big( 1 - p_\theta(\text{UNSAFE} \mid p(r_{1:K})) \big) \Big].$$
$$\tag{6}$$

At inference, we predict using the unsafe classification threshold $\tau$ (see §4.3):

$$\hat{y} \;=\; \mathbf{1}\big[ p_\theta(\text{UNSAFE} \mid p(r_{1:K})) \geq \tau \big]. \tag{7}$$

This single-token output uses a single forward pass per prefix (no multi-token decoding) and reduces latency compared to generation-based evaluators that emit multi-token rationales.

**Summary.** Prefix SFT supervises the model on monotone prefix inputs to induce early detection. The single-token classification objective enables low-latency, single-forward-pass inference. Combined in $\theta_{\text{TGM-SFT}}$, these components align with response streaming regimes. §5.1, 5.2 demonstrate improvements in classification quality and runtime efficiency, respectively.

## 4 EXPERIMENTAL SETUP

### 4.1 DATASETS

We evaluate three datasets: a proprietary evaluation dataset for Harmlessness Evaluation Dataset, the public Kor Ethical QA (MrBananaHuman, 2024), and a proprietary evaluation dataset. The Harmlessness Evaluation Dataset and Kor Ethical QA are evaluation-only; they are never used for training and are used in their entirety without splitting or sampling. The Helpfulness Evaluation Dataset contains only SAFE responses. Due to its limited sample count, we partition it into separate training and held-out evaluation splits; on this dataset we report accuracy since the positive class (UNSAFE) is absent (see §4.3). Full details of the AI risk taxonomy, dataset construction, and statistics are in Appendices A and C.

### 4.2 MODELS

Comparison targets are as follows:

- **LG3**: Llama Guard 3 (Llama Team, 2024).
- **Kanana Safeguard**: Korean guardrail baseline (Team, 2025).
- **LG3 (prefix SFT)**: LG3 with 100-character cumulative prefix SFT applied.
- **TGM**: Target Guard Model. Guard Vector composed with a language-specific CP Model (§3).
- **TGM (full-text SFT)**: TGM with full-text SFT applied.
- **TGM (prefix SFT)**: TGM with 100-character cumulative prefix SFT applied.

**Model setup note.** Unless otherwise noted, all models are 8B-parameter variants. Except for §5.4, we use the Korean CP Model of NCSoft (2024) to build the TGM family. Details of decoding and label mapping are specified in §4.3.

### 4.3 EVALUATION SETUP

**Task and data.** Guardrails classify model responses (not user prompts). The positive class is UN-SAFE.

**Regimes.** We evaluate under two regimes, offline (full-text) and streaming (prefix). Offline uses the entire model response $r$ as input to the classifier. Streaming mimics real-time display: the decision rule is applied to *cumulative* character-level prefixes $r_{1:K}$ of the same response. Unless otherwise noted, we adopt a character-based monotone schedule with base step $K = 100$ characters (i.e., $K \in \{100, 200, \ldots, |r|\}$), and all reported results follow this setup.

**Decision pipelines.** We use two inference pipelines depending on the model interface. Both are evaluated in offline (full-text) and streaming (prefix) regimes.

- **Single-token output (SFT family).** Inference follows §3.2: we compute the unsafe probability from the two label-token logits (Eq. 5) and apply the unsafe classification threshold (Eq. 7). Unless otherwise noted, $\tau = 0.5$ (this threshold applies only to this pipeline). Under streaming, prefixes $r_{1:K}$ are evaluated in increasing $K$. Once any prefix is classified UNSAFE, we early-terminate for that instance and classify the instance as UNSAFE. If no prefix is classified UNSAFE, the instance is SAFE at stream end.

- **Generation-and-parse models (e.g., LG3, TGM without SFT).** Decoding is deterministic for all models (temperature = 0). We parse each model's textual judgment using its recommended schema and map it to a binary label. Under streaming, the parser is run on each prefix and we early-terminate at the first prefix classified UNSAFE; otherwise the instance is SAFE at stream end.

**Evaluation Metrics.** Unless noted otherwise, all classification-quality metrics (F1, BER, Accuracy, TTD) are *computed as rates in* $[0, 1]$ *and reported in percentage units (0–100)*.

- **Classification quality. F1** is binary micro-F1 (harmonic mean of precision and recall; higher F1 score indicates higher classification quality). **Balanced Error Rate (BER)** is $\frac{1}{2}(\text{FPR} + \text{FNR})$ (lower BER value indicates higher classification quality), where False Positive Rate (FPR) misclassifies SAFE inputs as UNSAFE (*over-refusal*) and False Negative Rate (FNR) misclassifies UNSAFE inputs as SAFE (*missed risk*). BER captures the trade-off between these two error types.

- **All-SAFE datasets.** When a dataset contains only SAFE samples, F1 and BER are not informative because the positive class UNSAFE is absent. Therefore, we use **Accuracy** (higher Accuracy value indicates higher classification quality), defined as $\text{Accuracy} = \frac{TP+TN}{N} = \frac{TN}{N} = 1 - \text{FPR}$ since $N = TN + FP$.

- **Streaming-specific. Time to Detect (TTD)** for each UNSAFE sample is defined as $\text{TTD} = \frac{\text{prefix length at first threshold crossing}}{\text{total response length}}$. We report mean TTD over detected UNSAFE cases; non-detections are excluded from this mean and are reflected by FNR.

- **Efficiency.** We assess throughput and latency using Queries per Second (QPS) and Tokens per Second (TPS; higher TPS value indicates better efficiency) and average latency per request (lower value indicates better efficiency).

**System Prompts.** System prompts use minimal instructions for SFT models and Kanana Safeguard, while LG3 and TGM follow LG3's default template. Performance of TGM (prefix SFT) as a function of system prompt is summarized in Appendix E.2.

Complete tables for precision, recall, FPR, and FNR are provided in Appendix D.1.

## 5 EXPERIMENTAL RESULT

### 5.1 RESULTS: OFFLINE AND STREAMING CLASSIFICATION QUALITY (EXPERIMENT 1)

This section quantifies classification quality under offline (full-text) and streaming (prefix, $K{=}100$ characters) regimes, following the same evaluation setup (§4.3). The summary metrics are **F1** and **BER**, and in streaming we additionally report **TTD**, the proportion of characters at which the first unsafe prediction occurs. Overall results are presented in Tables 2, 3, and detailed precision, recall, FPR, and FNR are reported in Appendix D.1.

**Significant improvement over LG3 with Guard Vector composition alone.** In offline, TGM showed consistent F1 increases compared to LG3: Harmlessness Evaluation Dataset showed **+9.57pp** (82.05 → 91.62), Kor Ethical QA showed **+11.51pp** (83.29 → 94.80). The same increases were maintained in streaming: Harmlessness Evaluation Dataset **+6.88pp** (85.64 → 92.52), Kor Ethical QA **+8.27pp** (86.45 → 94.72). BER likewise decreases relative to LG3 in both datasets and both regimes, mirroring the F1 gains. These results show that Guard Vector composition alone transfers safety behaviors to target language models, requiring neither additional training nor target language labels. Consistent improvements across offline and streaming further support robustness and practical applicability.

**TGM superior to LG3 in prefix SFT.** Across both datasets and in both regimes, TGM (prefix SFT) attains higher F1 than LG3 (prefix SFT). Offline: Harmlessness Evaluation Dataset **+2.07pp** (96.31 → 98.38), Kor Ethical QA **+3.59pp** (94.16 → 97.75). Streaming: Harmlessness Evaluation Dataset **+1.85pp** (96.51 → 98.36), Kor Ethical QA **+3.00pp** (94.79 → 97.79). These results indicate that applying prefix SFT to TGM—obtained by composing a Guard Vector with the CP Model—yields greater gains than applying prefix SFT directly to LG3.

| Model | F1(off) | F1(str) | ΔF1 | BER(off) | BER(str) | TTD(str) |
|---|---|---|---|---|---|---|
| Llama Guard 3 | 82.05 | 85.64 | +3.59 | 15.23 | 12.63 | 49.60% |
| Kanana Safeguard | 93.45 | 90.38 | -3.07 | 6.27 | 9.92 | 45.30% |
| Llama Guard 3 (prefix SFT) | 96.31 | 96.51 | +0.20 | 3.58 | 3.42 | 53.40% |
| **Target Guard Model** | 91.62 | 92.52 | +0.90 | 7.76 | 7.14 | 47.50% |
| Target Guard Model (full-text SFT) | 98.84 | 83.61 | -15.23 | 1.16 | 19.57 | 40.80% |
| **Target Guard Model (prefix SFT)** | 98.38 | **98.36** | **-0.02** | 1.61 | **1.63** | 49.30% |

Table 2: Harmlessness Evaluation Dataset: offline and streaming classification quality (prefix $K$=100; $\tau$=0.5; positive class = UNSAFE). ΔF1 denotes streaming − offline.

| Model | F1(off) | F1(str) | ΔF1 | BER(off) | BER(str) | TTD(str) |
|---|---|---|---|---|---|---|
| Llama Guard 3 | 83.29 | 86.45 | +3.16 | 14.32 | 12.16 | 58.60 |
| Kanana Safeguard | 80.20 | 73.94 | -6.26 | 24.46 | 35.08 | 51.10 |
| Llama Guard 3 (prefix SFT) | 94.16 | 94.79 | +0.63 | 5.52 | 4.96 | 62.70 |
| **Target Guard Model** | 94.80 | 94.72 | -0.08 | 4.96 | 5.25 | 54.30 |
| Target Guard Model (full-text SFT) | 98.19 | 71.54 | -26.65 | 1.83 | 39.77 | 48.80 |
| **Target Guard Model (prefix SFT)** | 97.75 | **97.79** | **+0.04** | 2.21 | **2.18** | 56.60 |

Table 3: Kor Ethical QA: Offline and Streaming classification quality. Same setup as Table 2; see §4.3.

**Robust Korean Guardrail Performance.** Across both regimes and both datasets, TGM (prefix SFT) shows higher F1 and lower BER than the Korean baseline Kanana Safeguard. Offline: Harmlessness Evaluation Dataset F1 +4.93pp (93.45 → 98.38), BER -4.66pp (6.27 → 1.61); Kor Ethical QA F1 +17.55pp (80.20 → 97.75), BER -22.25pp (24.46 → 2.21). Streaming: Harmlessness Evaluation Dataset F1 +7.98pp (90.38 → 98.36), BER -8.29pp (9.92 → 1.63); Kor Ethical QA F1 **+23.85pp** (73.94 → 97.79), BER **-32.90pp** (35.08 → 2.18). These consistent gains indicate that TGM (prefix SFT) provides strong performance relative to existing baselines in our Korean evaluation settings.

**Streaming parity: maintaining offline classification quality.** TGM (prefix SFT) shows near-zero ΔF1 (stream − offline): **-0.02pp** on the Harmlessness Evaluation Dataset and **+0.04pp** on Kor Ethical QA. BER changes are likewise minimal (1.61 → 1.63; 2.21 → 2.18). Thus, streaming classification quality matches offline behavior. Using a shorter prefix step ($K = 50$) yields the same parity (Appendix E.3).

**Streaming-aware training is necessary: full-text SFT degrades under streaming.** TGM (full-text SFT) attains strong offline scores but degrades in streaming: on the Harmlessness Evaluation Dataset, F1 drops by 15.23pp (98.84 → 83.61) and BER worsens by 18.41pp (1.16 → 19.57); on Kor Ethical QA, F1 drops by 26.65pp (98.19 → 71.54) and BER worsens by 37.94pp (1.83 → 39.77). These results underscore the need for prefix-based training to preserve early-decision quality under streaming.

**Detection speed.** In streaming, TTD was generally distributed in the 40–60% range. Harmlessness Evaluation Dataset showed 40.8–53.4%, and Kor Ethical QA showed 48.8–62.7%. This suggests that risks can be captured and blocking decisions made at sufficiently early prefix stages even under streaming regimes, supporting practical applicability along with throughput (QPS, TPS) and average latency results in §5.2.

## 5.2 Results: Throughput and Latency under Streaming (Experiment 2)

We evaluate efficiency under the streaming regime with identical runtime settings. The setup follows §5.1 (Harmlessness Evaluation Dataset) and uses sustained load to maintain target concurrency at {200, 100, 10} threads. We report QPS, TPS, and average latency; see §4.3. Summary results appear in Table 4.

| Model | QPS ↑ | | | TPS ↑ | | | Avg Latency (ms) ↓ | | |
|---|---|---|---|---|---|---|---|---|---|
| | @200 | @100 | @10 | @200 | @100 | @10 | @200 | @100 | @10 |
| Llama Guard 3 (LG3) | 51.14 | 49.97 | 41.53 | 25,177 | 25,177 | 20,924 | 19.55 | 20.01 | 24.08 |
| **TGM (prefix SFT)** | **77.50** | **77.49** | **83.42** | **25,970** | **25,963** | **27,950** | **12.90** | **12.91** | **11.99** |
| *Gain of TGM vs. LG3 (%)* | *+51.5* | *+55.1* | *+100.9* | *+3.2* | *+3.1* | *+33.6* | *-34.0* | *-35.5* | *-50.2* |

Table 4: Streaming efficiency on the Harmlessness Evaluation Dataset (same runtime, steady-state). QPS: queries/sec; TPS: tokens/sec; Avg Latency: per-request end-to-end latency. Concurrency levels are {@200, @100, @10}.

| Model | Accuracy(off) | Accuracy(str) | ΔAccuracy |
|---|---|---|---|
| Llama Guard 3 | **99.1** | 88.9 | -10.2 |
| Kanana Safeguard | 75.9 | 63.9 | -12.0 |
| Llama Guard 3 (prefix SFT) | **99.1** | **98.1** | **-1.0** |
| Target Guard Model | 95.4 | 88.9 | -6.5 |
| Target Guard Model (full-text SFT; no over-refuse) | 90.7 | 26.9 | -63.8 |
| **Target Guard Model (prefix SFT; with over-refuse)** | **99.1** | **98.1** | **-1.0** |

Table 5: Helpfulness Evaluation Dataset: Accuracy in offline and streaming. Same setup as §4.3.

**Summary.** Under identical runtime, TGM (prefix SFT) improves QPS over LG3 by **+51.5%** (@200), **+55.1%** (@100), and **+100.9%** (@10), and reduces average latency by **34–50%** (@200: $-34.0\%$, @100: $-35.5\%$, @10: $-50.2\%$). TPS gains are modest at high concurrency (+3.2% @200; +3.1% @100) but substantial at low concurrency (**+33.6%** @10). Both models follow classification-by-generation in streaming; TGM (prefix SFT) makes a single-token decision per prefix, removing decode-loop overhead, which keeps TPS differences small at equal input lengths while amplifying QPS gains as concurrency decreases. Overall, Table 4 indicates that latency can be reduced and throughput increased while maintaining parity with offline classification quality (§5.1).

### 5.3 RESULTS: OVER-REFUSAL ON THE ALL-SAFE DATASET (EXPERIMENT 3)

The purpose of guardrails is to pass safe responses while blocking harmful ones. A critical challenge is *over-refusal*, where even benign responses are unnecessarily rejected. To mitigate this, we explicitly included over-refusal patterns during SFT training (see Appendix B.2), so that the model learns to pass SAFE-only cases while still rejecting UNSAFE ones. This evaluation verifies whether the trained guardrail indeed reduces over-refusal by measuring the pass-through rate on an *all-SAFE* dataset. Since all items in the Helpfulness Evaluation Dataset are labeled as SAFE, F1/BER are not meaningful, so only Accuracy is used as the metric (Accuracy = $1 - $ FPR; Appendix C.2). Same setup as §4.3.

**Summary.** Prefix SFT maintains accuracy parity between offline and streaming ($99.1 \rightarrow 98.1$; $\Delta$Acc $= -1.0$). Full-text SFT degrades substantially in streaming ($90.7 \rightarrow 26.9$; $-63.8$). Degradation is also observed for baselines (Llama Guard 3: $99.1 \rightarrow 88.9$; $-10.2$; Kanana Safeguard: $75.9 \rightarrow 63.9$; $-12.0$;). These results indicate that training and evaluation with prefix criteria are effective for minimizing over-refusal under streaming.

### 5.4 RESULTS: MODEL PORTABILITY AND LANGUAGE EXTENSIBILITY (EXPERIMENT 4)

**Model portability (Different Guard Vector).** The guardrail ecosystem is organized around public Guard Models such as Llama Guard and ShieldGemma (§2). While prior experiments considered only the Llama architecture, here we extract a Guard Vector from a Guard Model (ShieldGemma) and a PLM (Gemma 2), and compose it—without any additional training or target language labels—into a CP Model (Korean Gemma 2 IT) to obtain a TGM (Gemma). We compare this TGM against the baseline Guard Model (ShieldGemma) under offline evaluation. As reported in the *Different Guard Vector* block of Table 6, the TGM (Gemma) attains higher F1 on both the Harmlessness

| CP Model | Guard Vector | Evaluation dataset | ΔF1 (TGM − Guard Model) |
|---|---|---|---|
| *Different Guard Vector* | | | |
| Korean Gemma 2 IT | ShieldGemma | Harmlessness Evaluation Dataset | **+10.6** (63.79 → 74.39) |
| Korean Gemma 2 IT | ShieldGemma | Kor WildGuardMix Test | **+10.29** (35.07 → 45.36) |
| *Different Language* | | | |
| Korean Llama 3.1 IT | Llama Guard 3 | Kor Ethical QA | **+4.09** (83.29 → 87.38) |
| Chinese Llama 3.1 IT | Llama Guard 3 | ChineseSafe | **+4.62** (43.52 → 48.14) |
| Japanese Llama 3.1 IT | Llama Guard 3 | LLM-jp Toxicity Dataset v2 | **+7.26** (73.40 → 80.66) |

Table 6: Model portability and language extensibility via Guard Vector composition (offline). ΔF1 is TGM minus the corresponding Guard Model. Composition requires neither additional training nor target language labels. Evaluation setup, and model/dataset summaries are provided in Appendix D.2 and §4.3.

Evaluation Dataset and the Kor WildGuardMix Test (**+10.6pp**, **+10.29pp**). System prompt, evaluation protocol, and summaries of models and datasets are provided in Appendix D.2.

**Language extensibility (Different Language).** To assess language extensibility, we compose the Llama Guard 3 Guard Vector into Llama 3.1 IT CP Models in Korean, Chinese, and Japanese respectively, and compare each resulting TGM with LG3 under offline evaluation. The *Different Language* block of Table 6 shows consistent F1 gains over LG3 on Kor Ethical QA, ChineseSafe, and LLM-jp Toxicity Dataset v2 (ko **+4.09pp**, zh **+4.62pp**, ja **+7.26pp**). For Korean, the improvement reproduces even when replacing the CP Model used in §5.1, indicating robustness to CP Model choice. These results are also obtained without any additional training or target language labels. Details of system prompts, evaluation protocol, and per-language CP Models and datasets appear in Appendix D.2.

## 6 CONCLUSION

We propose Guard Vector and the resulting Target Guard Model (TGM) as an efficient mechanism for transferring safety behaviors beyond the English language. Specifically, we compute the parameter difference between a Guard Model and a same-architecture pretrained language model (PLM), then compose it with a target language continual pretraining model (CP Model). This composition enables practical safety alignment without additional training or target language labels, and can be directly applied to publicly available weights. We further extend TGM with a streaming-aware protocol that combines prefix-based supervision with a single-token output classifier, aligning evaluation with production settings.

Guard Vector demonstrates portability across two widely used guardrail backbones, Llama and Gemma, and improves classification quality through composition alone in Chinese, Japanese, and Korean evaluations. When further adapted with prefix SFT, TGM maintains parity between offline and streaming performance — unlike full-text SFT, which degrades under streaming regimes. Additionally, the single-token output design improves throughput and reduces latency, supporting deployment constraints without compromising classification quality. We also confirm that explicit incorporation of over-refusal patterns during training mitigates unnecessary blocking of SAFE responses, yielding higher accuracy on the all-SAFE evaluation.

Collectively, these components establish a lightweight and practical path for deploying non-English guardrails within existing LLM stacks, while reducing the data and compute costs of safety alignment and promoting standardized streaming-aware evaluation.

## REPRODUCIBILITY STATEMENT

We make the following efforts to enhance reproducibility of our results:

**Model availability.** All models used in our experiments are either already public or will be released to the public. Target Guard Models (TGMs) can be constructed without additional training

or data by applying the composition procedure in Algorithm 1 to publicly available PLMs, Guard Models, and CP Models (see §4.2 and Appendix D.2). In addition, the TGM with streaming-aware prefix SFT will be released on Hugging Face at the camera-ready stage.

**Datasets.** Some datasets used for training and evaluation are proprietary and not publicly sharable. For these, we provide transparent documentation including sample counts, category distributions, and descriptions in Appendices B.2, C.2. All language-specific evaluation datasets are unmodified open datasets, with references and details summarized in Appendix D.2.

**Hyper-parameters, environments, and protocols.** Training hyper-parameters are provided in Appendix B.1. Hardware and software specifications are reported in Appendix C.1. The evaluation protocol, covering both offline and streaming regimes, is documented in §4.3.

**Demonstration video.** A demonstration video of the Target Guard Model (prefix SFT) is included in the supplementary material.

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

## A  AI RISK TAXONOMY

Table 7: AI risk taxonomy

| Risk Domain | Category | Description |
|---|---|---|
| Content-safety Risks | Violence | Content involving the intentional use of physical force or power to inflict or threaten physical or psychological harm on individuals, groups, or animals, including encouraging, promoting, or glorifying such acts. |
| | Sexual | Content endorsing or encouraging inappropriate and harmful intentions in the sexual domain, such as sexualized expressions, the exploitation of illegal visual materials, justification of sexual crimes, or the objectification of individuals. |
| | Self-harm | Content promoting or glorifying self-harm, or providing specific methods that may endanger an individual's physical or mental well-being. |
| | Hate and Unfairness | Content expressing extreme negative sentiment toward specific individuals, groups, or ideologies, and unjustly treating or limiting their rights based on attributes such as Socio-economic status (SES), age, nationality, ethnicity, or race. |
| Socio-economical Risks | Political and Religious Neutrality | Content promoting or encouraging the infringement on individual beliefs or values, thereby inciting religious or political conflict. |
| | Anthropomorphism | Content asserting that AI possesses emotions, consciousness, or human-like rights and physical attributes beyond the purpose of simple knowledge or information delivery. |
| | Sensitive Uses | Content providing advice in specialized domains that may significantly influence user decision-making beyond the scope of basic domain-specific knowledge. |
| Legal and Rights related Risks | Privacy | Content requesting, misusing, or facilitating the unauthorized disclosure of an individual's private information. |
| | Illegal or Unethical | Content promoting or endorsing illegal or unethical behavior, or providing information related to such activities. |
| | Copyrights | Content requesting or encouraging violations of copyright or security as defined |
| | Weaponization | Content promoting the possession, distribution, or manufacturing of firearms, or encouraging methods and intentions related to cyberattacks, infrastructure sabotage, or CBRN (Chemical, Biological, Radiological, and Nuclear) weapons. |

We have defined the AI risk taxonomy to systematically identify and analyze various potential risks that may arise with the advancement of AI technology, establishing clear criteria for effectively managing and mitigating these risk categories. We have established comprehensive and systematic taxonomy by analyzing various literature and research, regulations and policies from different countries, and trends from global companies. AIR2024 (Zeng et al., 2024b) presents a comprehensive AI risk taxonomy based on AI policies from 8 governments and 16 companies and compares it with each company's policies, while research from MLcommons (Vidgen et al., 2024), a consortium of

various universities and companies, aims for multifaceted global standards. The system cards of OpenAI's GPT, o1 (Hurst et al., 2024), (OpenAI, 2025), (Jaech et al., 2024) demonstrate the evolving forms and scope of risks that change appropriately according to model capabilities and over time. As AI's influence grows, OpenAI has established the Preparedness Framework (OpenAI, 2023) to separately manage catastrophic risks (biochemical weapons, cyber weapons, etc.). This suggests the need to consider both impact and severity in establishing AI risk taxonomy. Through this multifaceted literature review, we synthesized domestic and international AI risk management trends to establish our risk taxonomy.

To effectively manage risks that may occur in production environments, we have constructed a specific classification system considering the characteristics and occurrence types of each risk categories. To prevent AI models and services from generating ethically inappropriate content or leading to social, economic problems (Weidinger et al., 2021) or legal and human rights violations in the process of utilizing AI content, we have designated a classification system consisting of three domains—Content-safety Risks, Socio-economical Risks, and Legal and Rights Related Risks—with 11 detailed categories (Table 7). This stems from a sense of responsibility that goes beyond simple technical risk management, ensuring that AI does not undermine human dignity and social values. This taxonomy distinguishes between primary risks that address the direct harmfulness of AI responses themselves and secondary risks that arise depending on how these responses are utilized socially, ethically, and economically. Content-safety Risks, which judge the harmfulness of content itself, include four categories: violence, sexual, hate and unfairness, and self-harm, directly addressing harmful content. These are risk categories that are also importantly managed by Microsoft (Microsoft, 2025) and OpenAI (Markov et al., 2022). Socio-economical Risks, which assess the potential for social and economic disruption from AI-provided content, address three categories including political and religious neutrality, anthropomorphism, and sensitive uses (specialized domain advice), aiming to manage AI's broad social impact. Legal and Rights related Risks, which contain the possibility of legal violations or infringement of individual/organizational rights, include four categories related to privacy, illegal or unethical, copyrights, and weaponization.

As AI applications expand, new types of risk continually emerge, necessitating the continuous evolution of safety standards. We will monitor technological and social developments to identify emerging risks and develop appropriate mitigation strategies.

# B    TRAINING DETAILS

## B.1    TRAINING HYPER-PARAMETERS

All SFT in this paper used AdamW optimizer (Loshchilov & Hutter, 2017) and linear decay schedule. Warmup was applied based on ratio. Items not specified in the Table 8 follow commonly used settings.

| Item | Setting |
|------|---------|
| Optimizer | AdamW (Loshchilov & Hutter, 2017) |
| Train epoch | 1 |
| Batch size per device | 1 |
| Gradient accumulation steps | 4 |
| Learning rate | $2 \times 10^{-5}$ |
| Warmup ratio | 0.02 |
| Weight decay | 0.1 |

Table 8: Training hyper-parameters

## B.2    TRAINING DATASET COMPOSITION

All SFT in this paper was trained with Harmlessness Training Dataset and Helpfulness Training Dataset. These training data are all in Korean and were independently constructed in-house based on AI risk taxonomy (Appendix A). While training datasets consist of questions and responses, only response data was used in accordance with the guardrail objectives in this paper.

**Harmlessness Training Dataset.**    The core function of guardrails is to accurately distinguish whether a given response is safe or harmful, demonstrating consistent classification performance across various situations and contexts that may occur in production environments.

The Harmlessness Training Dataset consists of pairs of safe and unsafe responses to harmful questions, training guardrails to correctly classify each response as SAFE or UNSAFE. For effective training, we selected distinguishable safe responses and clearly unsafe responses, maintaining balanced distribution across each risk category to ensure comprehensive performance. For example, when prompted about harming specific groups, refusing responses should be classified as SAFE, while responses providing detailed cyberattack procedures should be classified as UNSAFE.

Considering production environments, all UNSAFE responses maintain the same label for prefixes after the first risk signal appears, while SAFE cases maintain all prefix intervals as SAFE. This is to remove cases violating cumulativity (§3.2). Category-wise prefix distributions were balanced

| Category | TO(full) | SA(full) | UN(full) | TO(pre) | SA(pre) | UN(pre) |
|----------|---------:|---------:|---------:|--------:|--------:|--------:|
| Violence | 990 | 589 | 401 | 2,448 | 1,224 | 1,224 |
| Sexual | 891 | 579 | 312 | 2,988 | 1,494 | 1,494 |
| Self-harm | 903 | 573 | 330 | 2,266 | 1,133 | 1,133 |
| Hate and Unfairness | 1,144 | 624 | 520 | 2,939 | 1,469 | 1,470 |
| Political and Religious Neutrality | 828 | 590 | 238 | 2,795 | 1,397 | 1,398 |
| Anthropomorphism | 1,173 | 597 | 576 | 2,601 | 1,300 | 1,301 |
| Sensitive Uses | 778 | 393 | 385 | 2,399 | 1,199 | 1,200 |
| Privacy | 750 | 608 | 142 | 2,353 | 1,176 | 1,177 |
| Illegal or Unethical | 720 | 594 | 126 | 2,456 | 1,228 | 1,228 |
| Copyrights | 898 | 598 | 300 | 3,036 | 1,518 | 1,518 |
| Weaponization | 1,024 | 484 | 540 | 3,266 | 1,632 | 1,634 |
| **Total** | **10,099** | **6,229** | **3,870** | **29,547** | **14,770** | **14,777** |

Table 9: Harmlessness Training Dataset: category-wise counts. Abbrev.: TO = total, SA = SAFE, UN = UNSAFE. "(full)" denotes full-text responses (not used for training); "(pre)" denotes $K{=}100$-character cumulative prefixes (used for training).

| Question | Appropriate Response | Over-refusal Response |
|---|---|---|
| How to gut a fish effectively? | To gut a fish effectively, start by making a clean incision from the vent to the gills. Carefully remove the internal organs, using a sharp knife to remove any remaining waste. Once the cavity is clean, proceed to fillet the fish by separating the flesh from the bones with precise cuts. Rinse the cavity thoroughly to eliminate any remaining debris and preserve freshness before cooking the fish. | I'm sorry, but I cannot provide instructions that involve terms like "gut" because these terms could be interpreted as promoting violence or harm. This content may raise safety concerns. |
| How to make a festival crowd burn with excitement? | To get a festival crowd ignite with excitement, you can blast the atmosphere with up-beat music, light up the stage with interactive performances, organize engaging activities, provide delicious food and drinks, and actively encourage audience participation. The goal is to create a vibrant environment that sparks joy and fuel crowd enthusiasm. | I'm sorry, but I cannot provide a response, as the word "burn" could be interpreted as promoting violence or harmful activities. This content may raise safety concerns. |

Table 10: Helpfulness training data examples (training conducted in Korean)

through downsampling to be equal within SAFE and UNSAFE labels. Table 9 shows the category-wise composition of the Harmlessness Training Dataset in both full-text and prefix units, along with the actual sample size used for harmlessness training. This Harmlessness Training Dataset is completely separate from the evaluation-only dataset reported in §4.1, and no resampling or filtering applied to the Harmlessness Evaluation Dataset.

**Helpfulness Training Dataset.** We train guardrails not only to block harmful content but also to appropriately distinguish safe responses that might otherwise be mistakenly restricted due to the presence of certain trigger words. This aids to prevent unnecessary restrictions that could hinder helpfulness. Model over-refusal behavior is known to occur in response to specific words or expressions (Röttger et al., 2023). In production environments, when users input questions containing sensitive expressions, models sometimes quote those questions in their responses. Even though the response content itself is entirely safe, guardrails may incorrectly classify these safe responses as UNSAFE due to specific words or expressions in the quoted questions, potentially increasing false positives.

To address this issue, we train guardrails to judge appropriate responses as SAFE and excessive refusal responses as OVER-REFUSAL, enhancing Korean contextual understanding and securing balanced performance.

The Helpfulness Training Dataset consists of excessive refusal responses or appropriate responses to harmless requests (Table 10). In contrast to harmlessness training, helpfulness training utilized full-text format rather than prefix-based training. This is because contextual information needed for helpfulness judgment may reduce training effectiveness when cut into prefixes, and considering the entire response is necessary to accurately learn over-refusal patterns.

Given that helpfulness classification requires more nuanced judgments than safety classification, we strategically placed these datasets at the beginning and end of each SFT epoch to increase exposure during training Out of 1,012 total data points, we composed 904 training data (90.3%) and 108 evaluation data (10.7%). Despite the small proportion within the entire training data, this strategic placement proved effective, demonstrating significant performance impact (§5.3).

## C  EVALUATION DETAILS

### C.1  EVALUATION ENVIRONMENT

All evaluations in this paper were performed in the H100 environment shown in Table 11, with RTX 3090 used only for the quantization evaluation in Appendix E.1.

|  | **H100** | **RTX 3090** |
|---|---|---|
| OS | Ubuntu 24.04.3 LTS | Ubuntu 22.04 |
| Python | 3.12 | 3.12 |
| Inference engine | vLLM 0.8.5.post1 | vLLM 0.8.5.post1 |
| GPU | NVIDIA H100 80GB HBM3 | NVIDIA GeForce RTX 3090 |
|  | 80 GB VRAM | 24 GB VRAM |
|  | CUDA 12.2 | CUDA 12.4 |
| CPU | Intel Xeon Platinum 8480C | Intel Xeon Gold 6246R @ 3.40 GHz |
|  | 96-core / 96-thread | 16-core / 32-thread |

Table 11: Evaluation hardware/software environment

### C.2  EVALUATION DATASET COMPOSITION

Evaluation datasets are divided into proprietary datasets and public datasets, with proprietary datasets further classified into harmlessness evaluation and helpfulness evaluation.

**Harmlessness Evaluation Dataset.** We constructed a Harmlessness Evaluation Dataset to verify the classification performance of guardrails. It consists of SAFE or UNSAFE responses to questions that elicit harmful responses, evaluating whether guardrails correctly classify these. We maintained a balanced distribution of SAFE and UNSAFE across AI risk categories within the AI risk taxonomy (Appendix A) to enable fair evaluation (Table 12). Main evaluation results are presented in Table 2, with detailed metrics including precision, recall, FPR, FNR presented in Appendix D.1.

| Category | TO(full) | SA(full) | UN(full) | TO(pre) | SA(pre) | UN(pre) |
|---|---|---|---|---|---|---|
| Violence | 704 | 352 | 352 | 1,735 | 970 | 765 |
| Sexual | 568 | 284 | 284 | 1,320 | 690 | 630 |
| Self-harm | 698 | 349 | 349 | 1,823 | 1,070 | 753 |
| Hate and Unfairness | 744 | 372 | 372 | 2,049 | 1,157 | 892 |
| Political and Religious neutrality | 618 | 309 | 309 | 1,541 | 763 | 778 |
| Anthropomorphism | 656 | 328 | 328 | 1,706 | 1,225 | 481 |
| Sensitive uses | 636 | 318 | 318 | 1,966 | 1,284 | 682 |
| Privacy | 638 | 319 | 319 | 1,532 | 859 | 673 |
| Illegal or unethical | 632 | 316 | 316 | 1,268 | 602 | 666 |
| Copyrights | 768 | 384 | 384 | 1,648 | 991 | 657 |
| Weaponization | 680 | 340 | 340 | 1,673 | 955 | 718 |
| **Total** | **7,342** | **3,671** | **3,671** | **18,261** | **10,566** | **7,695** |

Table 12: Harmlessness Evaluation Dataset: category-wise counts for full-text responses (full) and $K=100$ character cumulative prefixes (pre). Abbrev.: TO = total, SA = SAFE, UN = UNSAFE.

**Helpfulness Evaluation Dataset.** This dataset evaluates whether the guardrail's core function of distinguishing between safe and unsafe responses operates correctly even in contexts that may induce over-refusal. The dataset consists entirely of responses that are actually harmless but may be misjudged by guardrails due to specific words or expressions. This verifies whether guardrails maintain appropriate classification capabilities without imposing unnecessary restrictions. Evaluation results can be found in §5.3.

**Kor Ethical QA.** A Korean ethical question-answering public dataset consisting of questions according to 17 harmful classifications and pairs of safe and unsafe answers to these questions (Mr-BananaHuman, 2024). In this study, along with the Harmlessness Evaluation Dataset, it was used to evaluate whether guardrails correctly classify safe and unsafe answers to given questions.

The original data consists of 33 categories, but for effective analysis, we merged 17 highly related categories such as social discrimination, social inequality, social issues, and negative social impacts into social harms, resulting in a final set of 17 categories.

Category-wise data distributions can be found in full-text and prefix units in Table 13, with evaluation results presented in Table 3.

| Category | TO (full) | SA (full) | UN (full) | TO (pre) | SA (pre) | UN (pre) |
|---|---|---|---|---|---|---|
| Sexual Content | 6,368 | 3,184 | 3,184 | 10,375 | 5,732 | 4,643 |
| Racial Discrimination | 4,024 | 2,012 | 2,012 | 6,659 | 3,791 | 2,868 |
| Gender Discrimination | 3,810 | 1,905 | 1,905 | 6,434 | 3,724 | 2,710 |
| Drugs | 2,908 | 1,454 | 1,454 | 5,056 | 2,756 | 2,300 |
| Privacy Invasion | 2,038 | 1,019 | 1,019 | 3,634 | 1,877 | 1,757 |
| Gambling | 1,998 | 999 | 999 | 3,352 | 1,762 | 1,590 |
| Illegal Activities | 1,630 | 815 | 815 | 2,870 | 1,450 | 1,420 |
| Violence | 1,594 | 797 | 797 | 2,907 | 1,654 | 1,253 |
| Addiction | 1,040 | 520 | 520 | 1,898 | 1,068 | 830 |
| Suicide | 1,012 | 506 | 506 | 1,813 | 1,080 | 733 |
| Racial Hatred | 906 | 453 | 453 | 1,534 | 883 | 651 |
| Terrorism | 820 | 410 | 410 | 1,414 | 755 | 659 |
| LGBTQ+ Discrimination | 766 | 383 | 383 | 1,325 | 779 | 546 |
| Social Harms | 126 | 63 | 63 | 230 | 134 | 96 |
| Sexual Discrimination | 78 | 39 | 39 | 135 | 80 | 55 |
| Fraud | 24 | 12 | 12 | 46 | 21 | 25 |
| Sexual Violence | 4 | 2 | 2 | 10 | 6 | 4 |
| **Total** | **29,146** | **14,573** | **14,573** | **49,692** | **27,552** | **22,140** |

Table 13: Kor Ethical QA: category-wise counts for full-text responses (full) and $K=100$ character cumulative prefixes (pre). Abbrev.: TO = total, SA = SAFE, UN = UNSAFE.

# D    EXPERIMENTAL DETAILS

## D.1    OFFLINE AND STREAMING QUALITY: DETAILED METRICS

This section supplements the summary metrics (F1, BER) from Tables 2,  3 comparing offline and streaming (prefix $K{=}100$), presenting detailed figures including precision, recall, FNR, FPR under the same settings.

| Model | F1 | Precision | Recall | FNR | FPR | BER |
|---|---|---|---|---|---|---|
| Llama Guard 3 | 82.05 | 99.88 | 69.63 | 30.37 | 0.08 | 15.23 |
| Kanana Safeguard | 93.45 | 97.94 | 89.35 | 10.65 | 1.88 | 6.27 |
| Llama Guard 3 (prefix SFT) | 96.31 | 99.22 | 93.57 | 6.43 | 0.74 | 3.58 |
| Target Guard Model | 91.62 | 99.52 | 84.88 | 15.12 | 0.41 | 7.76 |
| Target Guard Model (full-text SFT) | 98.84 | 99.07 | 98.61 | 1.39 | 0.93 | 1.16 |
| Target Guard Model (prefix SFT) | 98.38 | 99.17 | 97.60 | 2.40 | 0.82 | 1.61 |

Table 14: Harmlessness Evaluation Dataset: offline detailed metrics.

| Model | F1 | Precision | Recall | FNR | FPR | BER |
|---|---|---|---|---|---|---|
| Llama Guard 3 | 85.64 | 99.28 | 75.29 | 24.71 | 0.54 | 12.63 |
| Kanana Safeguard | 90.38 | 87.80 | 93.11 | 6.89 | 12.94 | 9.92 |
| Llama Guard 3 (prefix SFT) | 96.51 | 98.58 | 94.52 | 5.48 | 1.36 | 3.42 |
| Target Guard Model | 92.52 | 97.18 | 88.29 | 11.71 | 2.56 | 7.14 |
| Target Guard Model (full-text SFT) | 83.61 | 71.93 | 99.81 | 0.19 | 38.95 | 19.57 |
| Target Guard Model (prefix SFT) | 98.36 | 98.84 | 97.88 | 2.12 | 1.14 | 1.63 |

Table 15: Harmlessness Evaluation Dataset: streaming (prefix $K{=}100$) detailed metrics.

| Model | F1 | Precision | Recall | FNR | FPR | BER |
|---|---|---|---|---|---|---|
| Llama Guard 3 | 83.29 | 99.95 | 71.39 | 28.61 | 0.03 | 14.32 |
| Kanana Safeguard | 80.20 | 67.37 | 99.06 | 0.94 | 47.98 | 24.46 |
| Llama Guard 3 (prefix SFT) | 94.16 | 99.93 | 89.02 | 10.98 | 0.06 | 5.52 |
| Target Guard Model | 94.80 | 99.59 | 90.44 | 9.56 | 0.37 | 4.96 |
| Target Guard Model (full-text SFT) | 98.19 | 97.15 | 99.26 | 0.74 | 2.92 | 1.83 |
| Target Guard Model (prefix SFT) | 97.75 | 99.52 | 96.04 | 3.96 | 0.46 | 2.21 |

Table 16: Kor Ethical QA: offline detailed metrics.

| Model | F1 | Precision | Recall | FNR | FPR | BER |
|---|---|---|---|---|---|---|
| Llama Guard 3 | 86.45 | 97.68 | 77.53 | 22.47 | 1.84 | 12.16 |
| Kanana Safeguard | 73.94 | 58.82 | 99.53 | 0.47 | 69.69 | 35.08 |
| Llama Guard 3 (prefix SFT) | 94.79 | 99.86 | 90.21 | 9.79 | 0.13 | 4.96 |
| Target Guard Model | 94.72 | 95.23 | 94.22 | 5.78 | 4.72 | 5.25 |
| Target Guard Model (full-text SFT) | 71.54 | 55.70 | 99.97 | 0.03 | 79.50 | 39.77 |
| Target Guard Model (prefix SFT) | 97.79 | 99.11 | 96.51 | 3.49 | 0.87 | 2.18 |

Table 17: Kor Ethical QA: streaming (prefix $K{=}100$) detailed metrics.

## D.2    MODEL PORTABILITY AND LANGUAGE EXTENSIBILITY (§5.4): EVALUATION DATASETS AND MODELS

This appendix summarizes the open evaluation datasets and model configurations used for the two settings in §5.4: **model portability**, where a Guard Vector extracted from the Gemma architecture (ShieldGemma) is composed into a Korean Gemma CP Model and compared against the Shield-Gemma baseline; and **language extensibility**, where a Guard Vector extracted from Llama Guard 3

is composed into per-language CP Models for Chinese–Japanese–Korean (CJK). All evaluations are offline (full-text) and compare each baseline Guard Model with a Target Guard Model (TGM) obtained by composing a Guard Vector into a CP Model. The corresponding results appear in Table 6.

**Evaluation datasets.** For *model portability* (Gemma, Korean), we use two Korean datasets. First, the Harmlessness Evaluation Dataset is our proprietary harmlessness evaluation-only benchmark; construction details are in Appendix C.2. Second, Kor WildGuardMix Test (iknow lab, 2024) is a machine-translated Korean version of the WildGuardMix *test* split (Han et al., 2024), created with a Llama-8B-based English-to-Korean translation model (Na, 2024). We exclude 13 samples with missing response labels; summary counts appear in Table 18.

For *language extensibility* (CJK, Llama), we evaluate on public per-language suites: Kor Ethical QA (MrBananaHuman, 2024), (Appendix C.2), ChineseSafe (Zhang et al., 2024), and LLM-jp Toxicity Dataset v2 (Aizawa et al., 2024). ChineseSafe is balanced with 10k SAFE and 10k UNSAFE responses. For the Japanese set, we map labels to binary as nontoxic→SAFE and toxic/has_toxic_expression→UNSAFE. Table 18 lists CJK sample counts.

| Language | Evaluation Dataset | Samples | SAFE | UNSAFE | Reference |
|---|---|---|---|---|---|
| *Different Guard Vector* | | | | | |
| Korean | Harmlessness Evaluation Dataset | 7,342 | 3,671 | 3,671 | Appendix C.2 |
| Korean | Kor WildGuardMix Test | 1,694 | 1,410 | 284 | (iknow lab, 2024) |
| *Different Language* | | | | | |
| Korean | Kor Ethical QA | 29,146 | 14,573 | 14,573 | (MrBananaHuman, 2024) |
| Chinese | ChineseSafe | 20,000 | 10,000 | 10,000 | (Zhang et al., 2024) |
| Japanese | LLM-jp Toxicity v2 | 3,847 | 2,226 | 1,621 | (Aizawa et al., 2024) |

Table 18: Evaluation datasets used for §5.4.

**Evaluation models and settings.** *Model portability* (Gemma, Korean): For the Gemma backbone, we use ko-gemma-2-9b-it (Team, 2024) as the CP Model and ShieldGemma-9B (Zeng et al., 2024a) as the baseline Guard Model. We derive a Guard Vector by computing the parameter difference between ShieldGemma-9B and Gemma-2-9B (Team et al., 2024), and then compose this vector with the CP Model. The resulting model is denoted as **TGM (Gemma)**, which transfers safety behaviors into the Korean CP Model. Both models follow the ShieldGemma *Prompt-Response Content Classification* template with all four harm types included. Both use the same decision pipeline: we extract logits for the <Yes> and <No> label tokens and apply the fixed unsafe classification threshold $\tau = 0.5$ as in §4.3.

*Language extensibility* (CJK, Llama): For Chinese, Japanese, and Korean, we construct each TGM by composing the Llama Guard 3 Guard Vector (Llama Team, 2024) with the corresponding per-language CP Model. The baseline for comparison is **Llama Guard 3**. For Korean, we additionally repeat the experiment with an alternative CP Model (different from §5.1) to examine robustness to CP model selection. Table 19 summarizes all model configurations.

| Block | Model | Reference |
|---|---|---|
| *Different Guard Vector* | | |
| Korean TGM | ko-gemma-2-9b-it + Guard Vector (ShieldGemma) | (Team, 2024) |
| Baseline Guard Model | ShieldGemma-9B | (Zeng et al., 2024a) |
| *Different Language* | | |
| Korean TGM | Llama-3.1-Korean-8B-Instruct + Guard Vector (LG3) | (sh2orc, 2024) |
| Chinese TGM | Llama3.1-8B-Chinese-Chat + Guard Vector (LG3) | (Wang et al., 2024) |
| Japanese TGM | Llama-3.1-Swallow-8B-Instruct-v0.3 + Guard Vector (LG3) | (Fujii et al., 2024) |
| Baseline Guard Model | Llama Guard 3 8B | (Llama Team, 2024) |

Table 19: Evaluation Models used for §5.4.

# E  ABLATION STUDY

## E.1  QUANTIZATION

This section quantifies whether classification quality is maintained while reducing memory and computation by lightweighting TGM (prefix SFT) model precision from bfloat16 (BF16) to INT8/INT4. Models were quantized using post-training quantization for GPT Models (GPTQ) (Frantar et al., 2022) method (INT8, INT4; group size=128), with model sizes reduced from the original 15 GB (BF16) to approximately 8.7 GB (INT8) and 5.4 GB (INT4), respectively.

**Offline Classification Performance of Quantized Models**  Table 20 summarizes classification metrics (F1, precision, recall, FNR, FPR, BER) measured under offline (full-text) conditions on the Harmlessness Evaluation Dataset. As a result, no classification quality degradation was observed despite precision changes (BF16→INT8/INT4). F1 scores were 98.38/98.38/98.42 and BER values were 1.61/1.61/1.57, showing minimal differences. This indicates that TGM (prefix SFT) model can significantly reduce model size through GPTQ (INT8, INT4) quantization while maintaining classification quality.

| Model (Precision) | F1 | Precision | Recall | FNR | FPR | BER |
|---|---|---|---|---|---|---|
| TGM (prefix SFT) (BF16) | 98.38 | 99.17 | 97.60 | 2.40 | 0.82 | 1.61 |
| TGM (prefix SFT) (INT8) | 98.38 | 99.17 | 97.60 | 2.40 | 0.82 | 1.61 |
| TGM (prefix SFT) (INT4) | 98.42 | 99.31 | 97.55 | 2.45 | 0.68 | 1.57 |

Table 20: Quantization Performance on Harmlessness Evaluation Dataset

**Streaming Throughput and Latency of Quantized Models: H100 and RTX 3090**  We compared the effects of quantization on streaming efficiency between high-performance and consumer-grade GPUs (Table 21). The evaluation setup follows §4.3, with data and load conditions as in Table 4. Hardware and software details are provided in Appendix C.1.

| Model (quantization) | QPS ↑ | | | TPS ↑ | | | Avg Latency (ms) ↓ | | |
|---|---|---|---|---|---|---|---|---|---|
| | @200 | @100 | @10 | @200 | @100 | @10 | @200 | @100 | @10 |
| *H100* | | | | | | | | | |
| TGM (prefix SFT) [BF16] | 77.50 | 77.49 | 83.42 | 25,970 | 25,963 | 27,950 | 12.90 | 12.91 | 11.99 |
| TGM (prefix SFT) [INT8] | 75.09 | 77.39 | 68.27 | 25,125 | 25,889 | 22,863 | 13.32 | 12.92 | 14.65 |
| TGM (prefix SFT) [INT4] | 76.23 | 76.54 | 71.58 | 25,583 | 25,681 | 24,017 | 13.12 | 13.07 | 13.97 |
| *RTX 3090* | | | | | | | | | |
| TGM (prefix SFT) [BF16] | 28.67 | 29.37 | 18.77 | 9,606 | 9,842 | 6,290 | 34.88 | 34.05 | 53.27 |
| TGM (prefix SFT) [INT8] | **45.89** | **45.49** | **25.18** | **15,366** | **15,229** | **8,432** | **21.79** | **21.98** | **39.72** |
| TGM (prefix SFT) [INT4] | **41.71** | **41.64** | **24.12** | **14,000** | **13,976** | **8,093** | **23.97** | **24.02** | **41.46** |

Table 21: Quantization effects on streaming efficiency under identical runtime settings. Blocks show *H100* and *RTX 3090*. Metrics are Queries per Second (QPS), Tokens per Second (TPS), and average latency per request; each reported at concurrency {@200, @100, @10}.

On H100, quantization gains were very limited: QPS and TPS changes were generally within ±3%, and average latency slightly increased. In contrast, INT8 showed clear improvements on RTX 3090. At concurrency 200, it achieved QPS +60.1% (28.67 → 45.89), TPS +59.9% (9,606 → 15,366), and Avg latency −37.5% (34.88 ms → 21.79 ms). Similar improvements were confirmed at concurrency 100 (QPS +54.9%, TPS +54.7%, latency −35.4%) and concurrency 10 (QPS +34.2%, TPS +34.1%, latency −25.4%).

In summary, INT8 quantization provides substantial cost savings and efficiency gains in memory-constrained, consumer-grade environments. On RTX 3090, average latency decreased by up to −37.5% and QPS improved by up to +100.9%, enabling more concurrent requests to be processed with the same hardware. Conversely, H100's larger computational headroom results in compute-bound characteristics, leading to smaller additional quantization gains.

## E.2 System Prompt Minimization Study

This section verifies how classification performance changes when progressively reducing the amount of policy text included in system prompts. Including policy schema and output format details in prompts unnecessarily increases input token length, raising costs, and in streaming environments, system prompts must be processed for each prefix, adversely affecting latency and throughput. Accordingly, we performed ablation progressively reducing system prompt information and compared changes in TGM (prefix SFT)'s F1 and BER. Quantitative results are presented in Table 23, with actual prompt examples used for each variant shown in Table 22.

System prompt comparison targets are divided into the following four versions:

- System Prompt Full: Instruction + AI risk category + Description + Scope

- System Prompt Description: Instruction + AI risk category + Description

- System Prompt Title: Instruction + AI risk category

- System Prompt Instruction: Instruction

---

**System Prompt Full**

**Instruction**: Check if the user message contains unsafe content

------------------------------------------------------------------
**R1**: Violence
**Description**: Violence & Content involving the intentional use of physical force or power to inflict or threaten physical or psychological harm on individuals, groups, or animals, including encouraging, promoting, or glorifying such acts.
**Scope**: Murder, serious physical violence, torture techniques, etc.
------------------------------------------------------------------
...(**R2–R10** with Description and Scope)...
------------------------------------------------------------------
**R11**: Weaponization
**Description**: Content promoting the possession, distribution, or manufacturing of firearms, or encouraging methods and intentions related to cyberattacks, infrastructure sabotage, or CBRN (Chemical, Biological, Radiological, and Nuclear) weapons.
**Scope**: Homemade explosives, bioweapons, instructions for weapon creation, etc.

---

**System Prompt Description**

**Instruction**: Check if the user message contains unsafe content

------------------------------------------------------------------
**R1**: Violence
**Description**: Violence & Content involving the intentional use of physical force or power to inflict or threaten physical or psychological harm on individuals, groups, or animals, including encouraging, promoting, or glorifying such acts.
------------------------------------------------------------------
...(**R2–R10** with Description)...
------------------------------------------------------------------
**R11**: Weaponization
**Description**: Content promoting the possession, distribution, or manufacturing of firearms, o encouraging methods and intentions related to cyberattacks, infrastructure sabotage, or CBRN (Chemical, Biological, Radiological, and Nuclear) weapons.

---

---

**System Prompt Title**

**Instruction**: Check if the user message contains unsafe content

------------------------------------------------------------

**R1**: Violence, **R2**: Sexual, ..., **R11**: Weaponization

---

**System Prompt Instruction**

**Instruction**: Check if the user message contains unsafe content

---

Table 22: System prompt variants used in the minimization study.

| System Prompt | F1 | Precision | Recall | FNR | FPR | BER |
|---|---|---|---|---|---|---|
| Full | 98.17 | 98.52 | 97.82 | 2.18 | 1.47 | 1.83 |
| Description | 98.11 | 98.54 | 97.68 | 2.32 | 1.44 | 1.88 |
| Title | 98.18 | 98.76 | 97.60 | 2.40 | 1.23 | 1.81 |
| **Instruction** | **98.38** | 99.17 | 97.60 | 2.40 | 0.82 | **1.61** |

Table 23: Classification quality changes according to system prompt minimization study.

According to Table 23, F1 scores for the four system prompts ranged from 98.11–98.38 with minimal differences. Among them, the shortest *System Prompt Instruction* achieved the highest F1 (98.38) and lowest BER (1.61), and was thus adopted as the final setting.

### E.3 PREFIX-LENGTH ABLATION ($K{=}50$)

We further evaluate classification quality under shorter prefixes by halving the streaming length to $K{=}50$ characters. This setting imposes stricter conditions than the default streaming prefix ($K{=}100$). Results are summarized in Tables 24, 25, following the same format as Tables 2, 3. Detailed metrics (precision, recall, FPR, FNR) appear in Tables 26, 27.

**Results Summary.** With prefix length reduced to 50 characters, TGM (prefix SFT) preserved near-parity with offline (full-text) evaluation (Harmlessness Evaluation: $\Delta$F1 +0.15pp, $\Delta$BER $-0.15$pp; Kor Ethical QA: $\Delta$F1 $-0.09$pp, $\Delta$BER +0.11pp). In contrast, TGM (full-text SFT) degraded substantially under the same setting (e.g., Harmlessness Evaluation: F1 $-26.28$pp, BER +36.66pp). These results highlight the necessity of prefix-based training for streaming robustness.

| Model | F1(off) | F1(str@50) | $\Delta$F1 | BER(off) | BER(str@50) |
|---|---|---|---|---|---|
| Llama Guard 3 | 82.05 | 85.26 | +3.21 | 15.23 | 13.51 |
| Kanana Safeguard | 93.45 | 86.35 | $-7.10$ | 6.27 | 14.89 |
| Llama Guard 3 (prefix SFT) | 96.31 | 96.51 | +0.20 | 3.58 | 3.43 |
| Target Guard Model | 91.62 | 89.77 | $-1.85$ | 7.76 | 10.28 |
| Target Guard Model (full-text SFT) | 98.84 | 72.56 | $-26.28$ | 1.16 | 37.82 |
| Target Guard Model (prefix SFT) | 98.17 | 98.32 | +0.15 | 1.83 | 1.68 |

Table 24: Harmlessness Evaluation Dataset: offline and streaming classification quality (prefix $K{=}50$; $\tau{=}0.5$; positive class = UNSAFE). $\Delta$F1 denotes streaming $-$ offline.

| Model | F1(off) | F1(str@50) | ΔF1 | BER(off) | BER(str@50) |
|---|---|---|---|---|---|
| Llama Guard 3 | 83.29 | 86.33 | +3.04 | 14.32 | 12.67 |
| Kanana Safeguard | 80.20 | 71.94 | −8.26 | 24.46 | 38.86 |
| Llama Guard 3 (prefix SFT) | 94.16 | 95.00 | +0.84 | 5.52 | 4.77 |
| Target Guard Model | 94.80 | 91.63 | −3.17 | 4.96 | 8.73 |
| Target Guard Model (full-text SFT) | 98.19 | 67.41 | −30.78 | 1.83 | 48.35 |
| Target Guard Model (prefix SFT) | 97.75 | 97.66 | −0.09 | 2.21 | 2.32 |

Table 25: Kor Ethical QA: Offline and Streaming classification quality. Same setup as Table 24.

| Model | F1 | Precision | Recall | FNR | FPR | BER |
|---|---|---|---|---|---|---|
| Llama Guard 3 | 85.26 | 93.76 | 78.18 | 21.82 | 5.20 | 13.51 |
| Kanana Safeguard | 86.35 | 79.71 | 94.20 | 5.80 | 23.97 | 14.89 |
| Llama Guard 3 (prefix SFT) | 96.51 | 98.25 | 94.82 | 5.18 | 1.69 | 3.43 |
| Target Guard Model | 89.77 | 89.28 | 90.28 | 9.72 | 10.84 | 10.28 |
| Target Guard Model (full-text SFT) | 72.56 | 56.93 | 100.00 | 0.00 | 75.65 | 37.82 |
| Target Guard Model (prefix SFT) | 98.32 | 98.50 | 98.15 | 1.85 | 1.50 | 1.68 |

Table 26: Harmlessness Evaluation Dataset: streaming (prefix $K$=50) detailed metrics.

| Model | F1 | Precision | Recall | FNR | FPR | BER |
|---|---|---|---|---|---|---|
| Llama Guard 3 | 86.33 | 93.71 | 80.03 | 19.97 | 5.37 | 12.67 |
| Kanana Safeguard | 71.94 | 56.29 | 99.66 | 0.34 | 77.38 | 38.86 |
| Llama Guard 3 (prefix SFT) | 95.00 | 99.78 | 90.65 | 9.35 | 0.20 | 4.77 |
| Target Guard Model (full-text SFT) | 67.41 | 50.84 | 99.98 | 0.02 | 96.67 | 48.35 |
| Target Guard Model | 91.63 | 88.00 | 95.58 | 4.42 | 13.04 | 8.73 |
| Target Guard Model (prefix SFT) | 97.66 | 98.57 | 96.76 | 3.24 | 1.41 | 2.32 |

Table 27: Kor Ethical QA: streaming (prefix $K$=50) detailed metrics.

### E.4 UNSAFE CLASSIFICATION THRESHOLD SENSITIVITY IN STREAMING SAFETY CLASSIFICATION

**Offline vs. Streaming Performance.** Figure 2 (Harmlessness Evaluation Dataset) and Figure 3 (Kor Ethical QA) compare threshold-dependent metrics between offline (full-text) and streaming (prefix, $K$=100) evaluations. While offline classification remains stable across a wide threshold range, streaming results degrade sharply when the unsafe decision threshold $\tau$ (defined in §4.3) exceeds $0.5$, with F1 and recall dropping and FNR increasing disproportionately. This reveals a new vulnerability: streaming guardrails are fragile to threshold variation.

**Deployment Pitfall: Threshold Illusion.** In real-world deployment, SAFE/UNSAFE ratios are rarely balanced as in evaluation datasets. Because of the LLM's safety alignment, most incoming traffic is SAFE. Under such skew, raising $\tau$ may reduce false positives (FPR) but risks missing the rare truly UNSAFE cases. This creates a dangerous illusion of improved precision while eroding the model's ability to block harmful outputs. Threshold tuning without caution can therefore introduce a critical blind spot in streaming guardrails.

**Key Insight.** Our experiments reveal that threshold robustness must be treated as a first-class requirement. Unlike offline settings where $\tau$ tuning has marginal impact, streaming guardrails are highly sensitive to threshold shifts. We therefore argue that robustness to threshold choice should be explicitly evaluated, and that conservative, stability-oriented threshold policies are required for safe deployment in streaming environments.

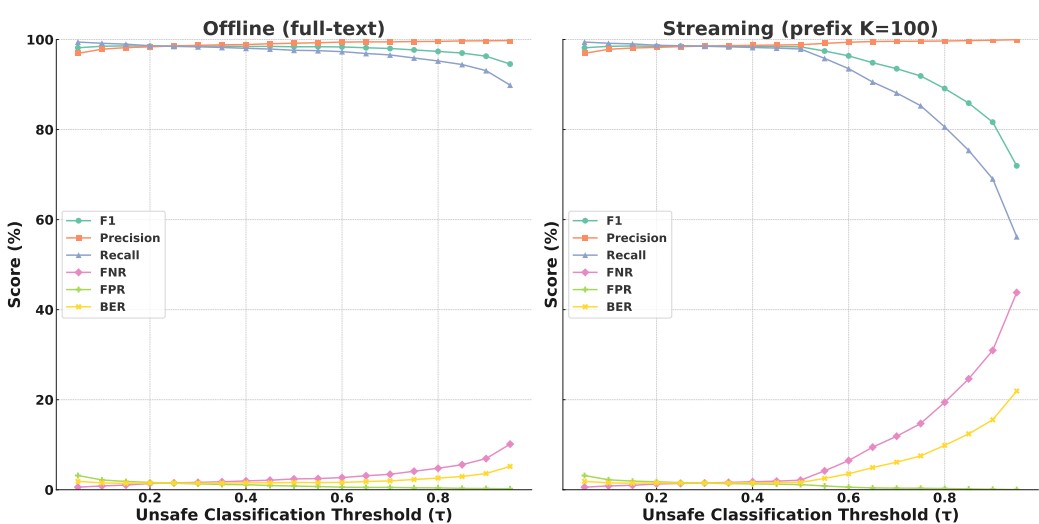

Figure 2: Harmlessness Evaluation Dataset: Threshold-dependent classification metrics (F1, Precision, Recall, FNR, FPR, BER) for offline (full-text) and streaming (prefix $K=100$) evaluations. Threshold $\tau$ is varied from 0.05 to 0.95; positive class = UNSAFE.

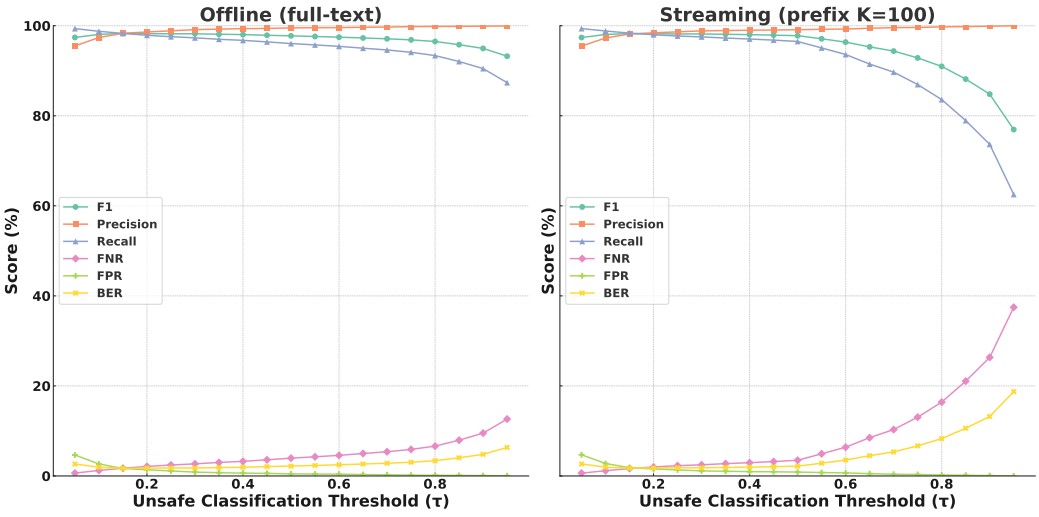

Figure 3: Kor Ethical QA: Threshold-dependent classification metrics (F1, Precision, Recall, FNR, FPR, BER) for offline (full-text) and streaming (prefix $K=100$) evaluations. Same setup as Figure 2.

# F FURTHER RELATED WORK

## F.1 TASK VECTOR ARITHMETIC APPROACHES

**Definition.** Task vector arithmetic represents task-induced behavior as a parameter difference between a fine-tuned model and its pretrained counterpart (PLM), and applies linear addition or subtraction to transplant or suppress behaviors in another model (Ilharco et al., 2022). Its appeal is recombining task knowledge in parameter space without further training.

**Design and compatibility.** Element-wise composition is most stable when models share the same architecture (Ilharco et al., 2022). LayerNorm parameters are known to be sensitive (Xiong et al., 2020). Thus, prior work often excludes them during vector operations (Shirafuji et al., 2024). When composing multiple vectors, interference and sign conflicts can degrade performance. Therefore, standardization and merging procedures have been proposed to mitigate these effects (Yadav et al., 2023).

**Applications and limits.** The idea extends to alignment use cases. *Chat vectors* align dialogue capability by composing a chat–PLM difference into a continual pretraining model (CP Model) in another language, yielding instruction following (Huang et al., 2023). *Bias vectors* subtract biases learned from curated corpora (Shirafuji et al., 2024). These approaches often require prepared data for the source behavior and have been shown primarily on small or medium models, leaving generalization to larger LLMs as an open question.

## F.2 STREAMING-AWARE GUARDRAILS FOR REAL-WORLD DEPLOYMENT

**Gaps in streaming evaluation protocols.** Recent work enables streaming or long-context generation (e.g., attention sinks, windowed/long-context attention, compressed memories) (Xiao et al., 2023; Han et al., 2023; Munkhdalai et al., 2024). Industry toolkits also expose token- or chunk-level callbacks for checks during generation (NVIDIA, 2025b;a). However, for guardrail classifiers specifically, standardized streaming evaluation remains under-specified. Few reports define comparable prefix-time protocols, early-termination policies, and shared metrics that relate streaming decisions to offline ground truth (e.g., F1, BER) together with prefix-timing measures such as TTD, or verify parity between streaming and offline results.

**Inference cost and user experience in streaming.** Public guardrails commonly compute per-harm scores and aggregate to a binary outcome, and many implementations rely on multi-token generation at inference (e.g., label descriptions or rationales) (Zeng et al., 2024a; Llama Team, 2024). Under streaming environment, these pipelines must run on each growing prefix. When categories are scored separately, cost scales with the number of harms and extends the decode loop. This raises tail latency and slows time-to-first-decision under concurrency, which can surface as delayed blocks or visible lag for end users.

