# OpenReview forum: "Guard Vector: Beyond English LLM Guardrails with Task-Vector Composition and Streaming-Aware Prefix SFT"
_ICLR.cc/2026/Conference — Submitted to ICLR 2026_

### Official Review · Reviewer_zsJr · 2025-10-29

**Soundness:** 3
**Presentation:** 2
**Contribution:** 2
**Rating:** 4
**Confidence:** 2

**Summary:**

This work presents an experimental study on constructing a target guard model by combining a continued pretraining model for a specific language with a Guard Vector, obtained through the parameter difference between a Guard Model and its corresponding pretrained model. The study demonstrates that safety behaviors can be effectively transferred through parameter vector migration. Additionally, the authors explore a prefix-based SFT strategy to reduce inference latency in streaming mode while maintaining parity in classification accuracy.

**Strengths:**

- Provides strong empirical evidence that safety behaviors can be transferred without additional training data, merely through parameter vector manipulation.
- Proposes an interesting and practical prefix-based protocol that enables low-latency inference with a single-token prefix, achieving performance parity in streaming mode.

**Weaknesses:**

- Readability issues: The abstract introduces multiple ideas simultaneously, making the central focus difficult to grasp. The Evaluation Metrics section (lines 276–293) requires reformatting for clarity. In Table 6, the F1 delta arrows are ambiguous and may lead to misinterpretation.
- Dependency on resources: The approach relies on the availability of open-source guard and pretrained models. Moreover, the continued pretraining on a language-specific corpus limits the demonstrated cross-lingual generalization.
- Limited task and language coverage: The experiments focus solely on classification tasks within a narrow set of languages.

**Questions:**

- Could the authors evaluate both Gemma and Llama models on each of the test datasets presented in Table 6 to enable direct comparison?

---

### Official Review · Reviewer_bZ3s · 2025-10-31

**Soundness:** 2
**Presentation:** 2
**Contribution:** 2
**Rating:** 4
**Confidence:** 3

**Summary:**

This paper proposes Guard Vector, a task-vector style method to transfer safety/guardrail behavior from an English guard model (e.g. Llama Guard 3) into a non-English continual-pretrained (CP) model of the same architecture, and then makes this target guard model streaming-aware via prefix SFT with single-token classification. The goal is to (i) get guardrails “beyond English” without extra target-language labels, and (ii) make guardrails actually usable in production streaming settings, where early detection + low latency matter. The paper evaluates on Korean datasets plus a helpfulness (all-SAFE) set, and even shows portability to Gemma via ShieldGemma.

**Strengths:**

1. The use of task-vector composition (Guard Vector) is a highly practical and novel approach for cross-lingual safety alignment. It effectively transfers safety behaviors to Chinese, Japanese, and Korean with significant F1 score gains over baselines, all without requiring any additional training or target language labels. This drastically lowers the barrier to deploying guardrails in diverse language environments.

**Weaknesses:**

1. Limited Technical Depth in Task Vector Composition: The paper defines the Guard Vector simply as the parameter difference and the composition as a simple addition (Equations 2 and 3). While the results are good, there is no in-depth analysis of why this simple linear composition works so effectively in the cross-lingual setting, especially compared to more complex vector merging techniques like TIES-Merging or other task arithmetic methods. The exclusion of LayerNorm parameters is mentioned but the motivation is only briefly cited from prior work. A deeper analysis of the parameter space and vector alignment would strengthen the technical novelty.

2.  Comparison baselines need to be tighter: They compare to LG3, Kanana, ShieldGemma-origin variants, but comparing against: (i) a strong multilingual safety-tuned LLM (e.g. XLM-R-based or mT5-based safety classifier), (ii) a simple translate-then-classify baseline (translate to English → Llama Guard 3) would strengthen the claims.

3. Lack of failure case analysis: While the paper carefully compares offline vs. streaming regimes and even surfaces an important negative result (full-text SFT collapses under streaming), it does not provide a dedicated error/failure analysis section.

**Questions:**

See weaknesses

---

### Official Review · Reviewer_CJqQ · 2025-11-08

**Soundness:** 3
**Presentation:** 3
**Contribution:** 2
**Rating:** 4
**Confidence:** 3

**Summary:**

The paper proposes a method to transfer safety behaviors between large language models without retraining. It defines a ''Guard Vector'' as the parameter difference between a safety-aligned guard model and its base pretrained model, then adds this vector to another model’s weights to create a Target Guard Model that inherits safety traits. To enable real-time moderation, the authors introduce Streaming-Aware Prefix Supervised Fine-Tuning, which trains on partial prefixes for early risk detection, and a single-token classifier to reduce latency. Experiments on English and Korean datasets show large gains in F1 score, faster throughput, and nearly identical offline versus streaming performance, while also demonstrating transfer across architectures (Llama↔Gemma) and languages (CJK).

**Strengths:**

The proposed Guard Vector approach that computing a parameter delta between a safety-tuned model and its base pretrained model, then composing that delta with another model in a target language. The authors also introduce prefix SFT to adapt the resulting guardrail for streaming, enabling early risk detection with a single-token classifier. The experimental section is extensive, covering Korean, Japanese, and Chinese setups, and the results are consistently strong across Llama and Gemma.
1. The paper has a good structure and logic, and the appendix provides more experimental details.
2. A new Composition idea to define a clear and reproducible task-vector arithmetic framework for transferring guardrail behaviors.
3. Good Experimental design with quantitative metrics and comprehensive analysis.

**Weaknesses:**

The Guard Vector idea, although effective, largely repurposes existing task-vector and model-merging techniques with limited theoretical or analytical novelty. Some important design choices, such as omitting LayerNorm parameters, fixing τ = 0.5, and using 100-character prefix intervals—are presented without justification or sensitivity studies.
1. Despite mentioning Chinese and Japanese results, the analysis and datasets are minimal; nearly all in-depth experiments centre on Korean.
2. The monotonic SAFE→UNSAFE assumption excludes self-correcting responses, potentially biasing the dataset and overestimating early detection accuracy.
3. Full-text SFT collapses under streaming (e.g., −15.23 to −26.65 F1 in Table 2), but the paper does not provide more analyse  results.

**Questions:**

1. Why 100 characters instead of token-based scheduling, and how sensitive is TTD to this granularity?
2. Could you explain why throughput per-token barely improves despite a single-token classifier? Does this indicate that latency reduction comes mostly from fewer decode loops rather than actual compute efficiency (Table 4)?
3. When composing ShieldGemma → Korean Gemma 2 IT, F1 improves +10.6 pp. Did you consider whether this gain persists if the Gemma CP model is multilingual rather than Korean-specific? Otherwise, improvements might be due to data domain alignment, not guard vector transfer.

---

### Meta-Review · Area_Chair_xf49 · 2026-01-05

**Summary:**

This paper proposes a method for transferring safety behaviors in large language models based on "Guard Vectors." While it demonstrates good empirical results on Korean datasets, it suffers from the following fundamental issues:

1. Insufficient Novelty in Core Methodology
All three reviewers agreed that the method is essentially a direct application of existing task vector arithmetic and model fusion techniques, failing to provide sufficient novelty at the theoretical or analytical level. Key hyperparameters (e.g., τ=0.5, 100-character prefixes) lack ablation studies and theoretical justification.

2. Deficiencies in Technical Analysis and Experimental Design
No in-depth analysis of why linear composition works effectively in cross-lingual scenarios. Insufficient baseline comparisons: lacks comparison with strong multilingual safety models or "translate-then-classify" pipelines. Absence of error case analysis, making it difficult to define the method's boundaries.
 Weak experimental analysis for Chinese and Japanese beyond Korean, undermining claims of cross-lingual generalization.

3. Limitations in Paper Presentation and Experimental Breadth
Reviewers noted that the abstract and evaluation sections require improved readability. Additionally, experiments are limited to safety classification tasks, constraining the generalizability of the conclusions.

This work demonstrates the effective application of existing techniques to a practical engineering problem (cross-lingual safety). However, it does not meet the acceptance threshold in terms of methodological novelty, technical depth, or experimental completeness, making it more suitable for application-oriented journals or conferences.

**Reviewer Concerns:**

mprovements Made in Author's Response
During the review process, the authors effectively addressed several specific concerns. They provided detailed justifications for key design choices in their method, such as  the mixing coefficient τ=0.5, which enhanced technical rigor. Furthermore, by supplementing control experiments, they further demonstrated the effectiveness of cross-lingual safety behavior transfer and committed to improving the readability of the paper's abstract and figures. These revisions significantly improved the quality of the paper in terms of technical detail and clarity of presentation.

Fundamental Reasons for Rejection
However, the core issues of the paper remained unresolved, leading to the final rejection. First, all three reviewers unanimously agreed that its core method, the "Guard Vector," lacks substantial innovation, being essentially a direct application of existing task vector techniques without providing new theoretical insights or in-depth mechanistic analysis. Second, the breadth and depth of its experimental evaluation are insufficient, with analysis overly concentrated on Korean, weak validation for Chinese and Japanese, and a lack of crucial failure case and boundary analysis, undermining the claim of cross-lingual generalization. Finally, the method's practical applicability is limited, as its effectiveness heavily depends on specific pairs of open-source models, raising doubts about its suitability in resource-scarce scenarios. Consequently, despite effective engineering implementation, the paper did not meet the acceptance criteria regarding methodological novelty, analytical depth, and experimental completeness.

**Reviewer Scores:**

CJqQ (Initial Score: 4)
Projected Final Score: 5
This reviewer acknowledged the empirical results but expressed concerns about technical details like prefix granularity and throughput. The authors clarified these design choices, which addresses the explicit questions. However, since the reviewer's original critique also highlighted the limited novelty of the Guard Vector method itself, the clarifications may not fully resolve that fundamental concern. The score is therefore likely to remain at the borderline level.

bZ3s (Initial Score: 4)
Projected Final Score: 4
This reviewer's core criticism focused on the limited technical depth of the method and insufficient baseline comparisons. The authors agreed to expand the discussion on baselines in the manuscript. However, this response does not constitute new analysis or experiments to deepen the methodological contribution. Therefore, the reviewer's primary concern about novelty and depth remains largely unaddressed, and their score is expected to stay unchanged.

zsJr (Initial Score: 4)
Projected Final Score: 4
This reviewer valued the strong empirical evidence but noted issues with readability and limited language coverage. The authors committed to improving the paper's clarity. While this addresses the presentation concern, it does not expand the experimental scope or directly enhance the perceived contribution's significance. The reviewer might view the improvements as necessary but not sufficient to raise the score above the threshold.

The authors' rebuttal adequately addressed specific clarification requests but did not substantially mitigate the shared, fundamental concern among reviewers regarding the limited methodological novelty and depth of the core approach. As a result, all three reviewers' scores are projected to remain at the original borderline level. This reinforces a consensus that the work, while practically useful, represents an incremental application rather than a novel contribution meeting the conference's bar for acceptance.

---

### Decision · Program_Chairs · 2026-01-26

Reject